# The Role of Macronutrients in the Pathogenesis, Prevention and Treatment of Non-Alcoholic Fatty Liver Disease (NAFLD) in the Paediatric Population—A Review

**DOI:** 10.3390/life12060839

**Published:** 2022-06-05

**Authors:** Thomas Pixner, Nathalie Stummer, Anna Maria Schneider, Andreas Lukas, Karin Gramlinger, Valérie Julian, David Thivel, Katharina Mörwald, Katharina Maruszczak, Harald Mangge, Julian Gomahr, Daniel Weghuber, Dieter Furthner

**Affiliations:** 1Department of Paediatric and Adolescent Medicine, Salzkammergutklinikum Voecklabruck, 4840 Voecklabruck, Austria; tom.pixner@gmx.at (T.P.); andreas.lukas@ooeg.at (A.L.); karin.gramlinger@ooeg.at (K.G.); dieter.furthner@ooeg.at (D.F.); 2Obesity Research Unit, Paracelsus Medical University, 5020 Salzburg, Austria; nathalie.stummer@stud.pmu.ac.at (N.S.); an.schneider@salk.at (A.M.S.); k.moerwald@salk.at (K.M.); k.maruszczak@stud.pmu.ac.at (K.M.); j.gomahr@salk.at (J.G.); 3Department of Paediatrics, Paracelsus Medical University, 5020 Salzburg, Austria; 4Department of Sport Medicine and Functional Explorations, Diet and Musculoskeletal Health Team, Human Nutrition Research Center (CRNH), INRA, University Teaching Hospital of Clermont-Ferrand, University of Clermont Auvergne, 63000 Clermont-Ferrand, France; vjulian@chu-clermontferrand.fr; 5Laboratory of Metabolic Adaptations to Exercise under Physiological and Pathological Conditions (AME2P), University of Clermont Auvergne, 63000 Clermont-Ferrand, France; david.thivel@uca.fr; 6Clinical Institute of Medical and Chemical Laboratory Diagnostics, Medical University of Graz, 8010 Graz, Austria; haral.mangge@medunigraz.at

**Keywords:** paediatric, non-alcoholic fatty liver disease, NAFLD, macronutrients, obesity, carbohydrates, fat, protein, nutrition, diet

## Abstract

Paediatric non-alcoholic fatty liver disease (NAFLD) has become the most common chronic liver disease in childhood. Obesity is the main risk factor. Nutrition and lifestyle are the key elements in preventing and treating NAFLD in the absence of approved drug therapy. Whilst recommendations and studies on macronutrients (carbohydrates, fat and protein) in adult NAFLD exist, the discussion of this topic in paediatric NAFLD remains contradictory. The purpose of this review is to provide state-of-the-art knowledge on the role of macronutrients in paediatric NAFLD regarding quality and quantity. PubMed was searched and original studies and review articles were included in this review. Fructose, sucrose, saturated fatty acids, trans-fatty acids and ω-6-fatty-acids are strongly associated with paediatric NAFLD. High consumption of fibre, diets with a low glycaemic index, mono-unsaturated-fatty-acids and ω-3-fatty-acids reduce the risk of childhood-onset NAFLD. Data regarding the role of dietary protein in NAFLD are contradictory. No single diet is superior in treating paediatric NAFLD, although the composition of macronutrients in the Mediterranean Diet appears beneficial. Moreover, the optimal proportions of total macronutrients in the diet of paediatric NAFLD patients are unknown. Maintaining a eucaloric diet and avoiding saturated fatty acids, simple sugars (mainly fructose) and a high-caloric Western Diet are supported by literature.

## 1. Introduction

Non-alcoholic fatty liver disease (NAFLD) has become the most prevalent type of chronic liver disease in the paediatric population. NAFLD affects 3–10% of the general paediatric population and up to 40–50% of children and adolescents with obesity or overweightness, and is expected to become the leading cause of liver failure in the Western world under the age of 18 years old [1,2,3,4]. Its prevalence in studies varies depending on the diagnostic methods used, ethnicity, age, gender, geographical region and the population sampled [5]. NAFLD encompasses a disease spectrum ranging from hepatic steatosis (at least 5% of macro vesicular fat present in the liver) to non-alcoholic steatohepatitis (NASH) and hepatic fibrosis. It is traditionally defined by exclusion criteria, as morphological changes need to happen in the absence of significant alcohol consumption, genetic/metabolic disorders, infectious disease of the liver and steatogenic medications [6]. The diagnosis of NAFLD before the age of 3 years old requires reinvestigation for an underlying metabolic disease [7,8,9]. It must be noted that NAFLD is also diagnosed in normal-weight children (lean-NAFLD), with its prevalence ranging from 1.5–5% [3,10,11].

Obesity remains the most important risk factor for NAFLD [10,12], favouring metabolic impairments (e.g., insulin resistance, dyslipidaemia and impaired glucose tolerance) that affect the liver and cause NAFLD as early as childhood. Insulin resistance leads to increased gluconeogenesis and accumulation of hepatic fat. Hyperglycaemia promotes lipid accumulation in hepatocytes by stimulating lipogenesis [13,14], promoting the release of pro-inflammatory cytokines [15]. The process is bidirectional, as NAFLD in turn aggravates the inflammatory response in obesity [16,17]. NAFLD is considered the hepatic feature of obesity and metabolic syndrome [18,19]. Paediatric NAFLD is thus associated with an increased risk of disturbed glucose metabolism, arterial hypertension, renal dysfunction and chronic kidney disease, decreased bone mineral density, obstructive sleep apnoea (OSAS) and polycystic ovary syndrome (PCOS) [20,21,22,23,24,25,26]. In adults, NAFLD increases the risk of cardiovascular events eightfold, or threefold for type 2 diabetes, and it is a severe risk factor for hepatocellular carcinoma [27,28,29]. US children and adolescents first diagnosed with NAFLD frequently display features of NASH and advanced fibrosis has been present in in 10–25% of these cases [30]. Children with NAFLD display a different histological pattern of the liver compared with adults, with evidence for a more aggressive disease phenotype [31,32,33]. Abdominal visceral obesity in particular correlates with histological damage and hepatic fibrosis in children [34].

NAFLD has been traditionally presented as a mere imbalance of too much caloric intake and insufficient physical activity. The over-consumption of food, especially that which is rich in total and saturated fats and refined sugar, is considered a major risk factor for the development of NAFLD and obesity. Children with NAFLD (biopsy-proven) have been reported to more likely consume a Western diet, but also to have a more sedentary lifestyle [35,36]. While normal-weight children have been shown to dedicate about 16 h per week to sedentary activities (e.g., reading, watching TV or playing video games and drawing), this time reaches about 25 h per week in children with NAFLD and overweightness [37,38].

In addition to behavioural factors, a number of genetic variants have been identified and associated with both paediatric and adult NAFLD. The variants in patatin-like phospholipase domain-containing-protein-3 (PNPLA3) are linked to progression to NASH and fibrosis. PNPLA3′s gene expression is also directly related to the nutritional status. It is upregulated by over-nutrition and downregulated in a fasting state [39,40,41,42,43]. Mutations in the transmembrane-6 superfamily member 2 (TM6SF2) affect lipid secretion, variations in the glucokinase regulator (GCKR) affect monosaccharide uptake and lipid synthesis and MBOAT7 affects lipid metabolism. Other SNPs are known to affect mitochondrial functioning, transmembrane transport of sugars and bile acids and modify inflammatory pathways [16,44,45]. The influence of genetic variants on NAFLD is supported by prevalence variations among different ethnic groups, family clusters and the occurrence of lean NAFLD, as well as variable progression of liver disease in certain individuals with known variants [43,46].

Epigenetic factors (e.g., in utero development and maternal factors) are also considered as risk factors. They determine the risk of disease, not only in childhood, but also later in life [47]. Antenatal risk factors include hyperglycaemia in utero and gestational diabetes, maternal obesity and fructose intake, excessive weight gain during pregnancy, metabolic syndrome during pregnancy, caesarean section, low and high birth weights, absence of breast feeding and early exposure to antibiotics [48,49,50,51,52,53,54].

Due to a lack of pharmacological options for the treatment of paediatric NAFLD, nutritional intervention and lifestyle modification remain recommended as a first-line treatment. The focus of nutritional intervention is on lowering energy intake and healthier eating profiles, while lifestyle aims at increasing physical activity (PA) and reducing sedentary behaviour [55,56]. The majority of studies on interventions in paediatric NAFLD focus on the combination of nutrition and lifestyle [57].

While there is a multitude of studies and recommendations for nutritional intervention in adult NAFLD, the optimal quantity and quality of macronutrients (carbohydrates, fat, protein) in paediatric NAFLD remains debated [58,59,60,61,62]. The current narrative review, therefore, aims to provide insight into what is known regarding the role of macronutrients in the development and nutritional therapy of paediatric NAFLD. It does not discuss nutritional supplements and vitamins, but solely macronutrients themselves and in the context of diets/dietary patterns.

## 2. Materials and Methods

We performed a literature search for this review using the electronic database PubMed. As this is a narrative review, no systematic criteria (e.g., PRISMA) were applied. The search parameters included original articles or reviews about patients (any sex, age, race or comorbidity were included). The first search without filters was performed using the terms ‘‘NAFLD” and ‘‘carbohydrates” and ‘‘fat” and ‘‘protein” for publications until 31 December 2021. In an additional search, the parameters “NAFLD” and “macronutrients” (articles up to and including 31 December 2021) were searched for. Emphasis was put on the paediatric population. For the purpose of this review, the paediatric population was defined as being under the age of 18 years old. Secondary literature (references from studies and reviews) was also reviewed.

## 3. Results

### 3.1. General Findings

The first literature search resulted in 1049 articles and the second search resulted in 754 publications. Two of the authors (TP and DF) reviewed the articles. After screening the titles and abstracts, a total of 361 articles of relevance remained. Of these, 283 were used to write this narrative review.

At present, there is no specific diet that is universally recommended for nutritional therapy in paediatric patients with NAFLD. Overall, most studies and societal guidelines give general recommendations, mentioning a low-caloric healthy diet (low in carbohydrates, fructose, saturated fatty acids and trans-fatty acids, but prioritising low-glycaemic-index foods) [63,64]. With regards to lifestyle interventions and weight reduction, no systematic difference is made between children and adolescents with obesity and/or NAFLD [65,66].

The American Heart Association suggests a balanced diet consisting of vegetables, fruits, low saturated fats, whole grain and fish, with low fructose and salt intake [67,68]. While some authors state that no diet is clearly superior, some advocate for the Mediterranean Diet [57,69,70,71,72,73]. Avoiding the overconsumption of the previously mentioned macronutrients and therefore the resulting obesity is a key point in preventing and treating NAFLD [74]. No author or guideline mentioned a concrete caloric restriction, specific meal composition, meal frequencies and distribution of macronutrients (e.g., protein in the evening and carbohydrates in the morning) or a specific recommendation that differed from general healthy diet recommendations. Dietary recommendations are generally observation based and there has been only a small number of randomised controlled trials of diet as a monotherapeutic approach for paediatric NAFLD [57,68,75].

As most patients with NAFLD suffer from obesity or overweightness, caloric restriction is a consistent recommendation for the treatment of adults and children with NAFLD. However, some controversy remains regarding the role of caloric restriction in treating paediatric NAFLD [76]. Mouzaki et al. argue that it is difficult to clarify the role of a specific macronutrient vs. the total caloric intake [69]. Eslamparast et al. state that caloric restriction is the most important element in the dietary treatment of NAFLD. Special emphasis is put on the avoidance of high amounts of carbohydrates, simple sugars, fat and protein, as well as low fibre intake [74].

A general point of agreement is that the quality of the macronutrient is essential, as is an individualised approach to therapy. A poor-quality diet (e.g., convenience foods and those high in saturated fat and sugars) may even cause NAFLD in the absence of obesity [74,75,77,78]. Panera argues that the composition of macro- and micronutrients is more important in the pathogenesis and treatment of NAFLD than the total energy intake [43]. Another point of agreement is that nutritional intervention should be paired with physical exercise and the reduction of sedentary behaviour [57,67]. Recent studies suggest that a reduction in the intake of simple carbohydrates (especially high fructose/sucrose) and saturated fats benefits the prevention of paediatric NAFLD [46,63,74]. There is evidence for a metabolic threshold. This describes a relative body fat content that needs to be achieved before high amounts of fructose and other high-caloric macronutrients lead to hepatic and metabolic impairment [79]. The consumption of food enriched in fat and fructose may alter the gut microbiome and the intestinal barrier function, thus leading to endotoxemia and low-grade inflammatory processes [43,80,81].

### 3.2. Macronutrients

Although general recommendations for macronutrients exist (see above), the influence of macronutrient intake and composition in paediatric NAFLD remains a topic of major interest.

Vos et al. stated that there was no significant difference between the fraction of calories consumed from fat, carbohydrates and protein in children with steatosis and NASH [82]. Anderson et al. reported inconsistent associations between absolute macronutrient intake and liver outcomes. They found evidence neither for a particular macronutrient, nor that intake at a particular age was strongly associated with hepatic outcomes [83]. Similarly, Gibson et al. found no significant differences in macro- and micronutrient intake between biopsy-proven paediatric NAFLD and obese controls. Therefore, they recommended that paediatric NAFLD patients should follow general paediatric recommendations for weight loss and not specific guidelines [84]. These three studies contradict general findings focusing on fat and carbohydrate reduction (especially fructose) as primary targets in nutritional intervention. In the following paragraphs, each macronutrient is reviewed on its own.

#### 3.2.1. Carbohydrates

##### Carbohydrates—General Recommendations

The World Health Organization (WHO) currently recommends a reduction of free sugar (primarily sucrose and fructose) intake to less than 10% daily of the total energy intake (see Table 1 for WHO recommendations) [85]. In general, the majority of calories consumed are provided by carbohydrates. The intake of total energy and carbohydrates are higher in children and adolescents with NAFLD than in healthy children [86]. Carbohydrates can be grouped into sugars—monosaccharides, disaccharides, oligosaccharides and polysaccharides—and sugar alcohols. While several carbohydrates have been associated with NAFLD, the primary focus is on fructose [69,87]. In addition, added sugar that stems from glucose and fructose is pivotal to NAFLD development [86]. For example, in US adolescents, average consumption of 94 g of added sugar daily, equalling 17.9% of energy intake, has been reported [53,54,88,89,90].

##### Carbohydrates in the Pathogenesis of Paediatric NAFLD

A diet rich in carbohydrates provides the major source for hepatic free-fatty-acid (FFAs) synthesis. In children with NAFLD, the production of FFAs from carbohydrates contributes 30% of the total FFAs, whereas in healthy children, high carbohydrate intake produces only 5% of total FFAs [95]. The effect of carbohydrate conversion to triglycerides and development of NAFLD has been more closely associated with fructose and sucrose than with glucose [96].

Additional glucose promotes lipid synthesis via the activation of the transcription factor carbohydrate-responsive element-binding protein (ChREBP). Dietary glucose and fructose have different effects on mitochondrial protein acetylation and malonyl-CoA. This results in a greater reduction in fatty acid (FA) oxidation and greater lipid synthesis in response to the extra-fructose diet compared with the extra-glucose diet [97,98,99].

##### Carbohydrates in Treatment and Prevention of Paediatric NAFLD

Low-carbohydrate diets may improve weight loss and reduces hepatic steatosis, as well as reduce triglycerides (TGs) and increase the HDL levels [100,101,102,103]. In a 2019 study, Schwimmer et al. investigated 11–16-year-old boys with NAFLD (≥10% liver fat content). The boys were assigned to either a low-sugar diet or a regular control diet. The study aimed to provide less than 3% of their daily energy from added sugars (intervention). During the study, the low-sugar diet reduced the liver fat content by 8% (compared with 1% in the control diet group). No significant changes in TGs or fasting insulin were noted, but there was an improvement in plasmatic liver enzymes [104]. In a 2020 study, Goss et al. reduced the liver fat content by 6% via a carbohydrate-reduced diet over the course of 8 weeks. In this study, girls and boys with obesity and NAFLD were assigned to either a low-carbohydrate or a low-fat diet. The low-fat diet resulted in a reduction in liver fat of 1%. The study demonstrated carbohydrate restriction as being superior to a low-fat diet when the daily caloric requirement was maintained, not only resulting in a reduction in the hepatic fat content, but also an improvement in insulin resistance [76].

Both studies provide evidence that a general reduction in carbohydrates can result in a reduction in the hepatic fat content and improvement in metabolic parameters.

##### Fructose and Sucrose—Fructose and Sucrose in the Epidemiology and Pathogenesis of Paediatric NAFLD

The consumption of fructose has increased over the past several hundred years and dramatically over the past few decades in parallel with the consumption of sucrose and high-fructose corn syrup (HFCS) [91]. Since the 1980s, an increase as high as 300% over a period of 20 years has been recorded. Studies claim that fructose may be as addictive as certain drugs [87,105,106]. The consumption of fructose and sucrose has equalled the epidemic rise in paediatric obesity and NAFLD [69,91,107,108,109,110,111,112,113].

In its natural form, fructose and sucrose can be found in fruits, vegetables and a great number of foods and beverages. Fructose, sucrose and HFCS can also be found in large quantities in processed foods and sugar-sweetened beverages. This is especially relevant as children and adolescents with NAFLD consume significantly more soft drinks than the general population [114]. NAFLD patients generally consume nearly two to three times more fructose (365 kcal) than healthy controls (170 kcal) [38]. It is relevant that the majority of fructose does not come from fruits or vegetables in paediatric NAFLD patients, but from processed foods. In a collective of 8–16-year-olds, only 25% of NAFLD patients consumed one or more fruits per day. In normal-weight children, daily fruit consumption was the case in 64.7%, and in 45% of children with obesity but without NAFLD. In contrast, an epidemiological study from Finland found an inverse relationship between fructose intake and NAFLD. However, in this study, the majority of fructose was consumed through fruit intake. Fruits contain less fructose than soft drinks and processed foods, but also contain constituents (flavanols, epicatechin, ascorbate and other antioxidants) that may antagonise the negative metabolic and inflammatory effects of fructose [115,116].

In 2014, Sullivan et al. demonstrated that the absorption and metabolisation of fructose is more effective in children with NAFLD than in lean, healthy children, and also resulted in an exacerbated metabolic profile [117]. Fructose and sucrose were found to interact with transcriptional factors affecting the gene expression involved in glycolysis and lipogenesis [118].

Studies have demonstrated that fructose is associated with increased insulin resistance, leptin resistance and a decrease in adiponectin (the latter two being hormones produced by adipocytes) [69,110,111]. Leptin resistance is induced by chronic fructose consumption. Even before weight gain, it aggravates high-fat-induced obesity. This effect is fully reversible when fructose is eliminated from the diet. Leptin is an anorexigenic hormone; hence, in leptin resistance, satiety cannot be achieved [110,111]. Adiponectin is decreased in patients with obesity and involved in lipid and carbohydrate metabolism. High-fructose diets also reduce the activity of PPARα and hepatic lipid oxidation, and stimulate NF-kB expression. In animal studies, this has led to oxidative stress, as well as hepatic steatosis and hepatic fibrosis. Fructose is absorbed in the small intestine via the GLUT5 transporter and is taken up by the liver, testes and other organs. (Over)consumption of a fructose-rich diet induces de novo lipogenesis, causing elevated triglyceride and cholesterol levels with dyslipidaemia [119].

Whether fructose consumption induces gut leakiness and intestinal dysbiosis remains under discussion [120]. Sullivan et al. suggested altered intestinal bacterial flora fermentation (fructose into hydrogen) or an up-regulation of the GLUT5 transporter in the intestinal epithelium to be at the root of dysbiosis [117]. Jones at al. reported that a high-fructose diet was associated with lower abundance of beneficial microbes (Eubacterium and Streptococcus), which are involved in carbohydrate metabolism, aiding dysbiosis [121]. A 2020 study by Aleman et al. demonstrated that a daily dose of 75 g of fructose in obese patients did not cause increased gut permeability or endotoxemia [122]. This contradicts previous reports that stated that fructose causes bacterial overgrowth, an increase in intestinal permeability, precipitating endotoxemia and, hence, the initiation of an inflammatory cascade driving the progression of NAFLD [123,124].

It has been reported that regular consumption of fructose or sucrose in a caloric-stable diet increases the synthesis of fatty acid synthesis, an effect that is not present with glucose [125]. Studies on isocaloric diets using sucrose (glucose–fructose disaccharide) or a 50:50 mixture of glucose and fructose showed that the mixture of monosaccharides induced slightly more steatosis of the liver [126]. Studies have also shown the additive effect of a diet high in fat and sucrose on the development of NAFLD [127].

##### Fructose and Sucrose—Fructose and Sucrose in the Treatment and Prevention of Paediatric NAFLD

In nutritional studies, it has been shown that lowering fructose, sucrose and HFCS intake may be protective against the development of NAFLD in children with obesity. As sucrose is 50% fructose and 50% glucose, it must be considered that a reduced intake of sucrose also results in a reduced intake of fructose. Studies cited in this review specifically listed fructose or sucrose. The amount of fructose was shown to be more relevant than the total amount of carbohydrates in general, and the amount of fructose consumed at the age of 14 years old was a predictor of the presence of NAFLD at the age of 17 years [128]. Studies reported that it normally takes at least 8–24 weeks of a high fructose-diet to develop NAFLD [129]. Fructose-induced NAFLD is also able to develop in the absence of weight gain [130]. Even short-term interventions of nine days of isocaloric fructose restriction could show decreases in liver fat, visceral adipose tissue and de-novo lipogenesis, as well as improved insulin profiles [131]. The FRAGILE study was able to demonstrate that a decrease in fructose intake reduces markers of liver dysfunction in children and adolescents with NAFLD [79]. In a study by Jin et al. during a 4-week randomised controlled double-blinded beverage intervention study, it was reported that reduction in dietary fructose in Hispanic American adolescents with NAFLD improved several plasma markers associated with metabolic syndrome (e.g., adipose insulin sensitivity, high-sensitivity C-reactive protein and low-density lipoprotein oxidation) [132].

In healthy subjects, trials of a 4-week moderate (1.5 g/kg/day) and 7-day high-fructose (3.5 g/kg/day) supplementation demonstrated the negative effects of additional fructose. In the 7-day trial, dyslipidaemia and ectopic lipid deposition in the liver and muscles occurred, while the moderate 4-week modification led to an increase in the plasmatic TG and glucose levels [133,134]. This is in accordance with adult data, showing that even shorter periods (6 days) are sufficient to induce dyslipidaemia, as well as hepatic and adipose tissue insulin resistance in healthy subjects with a high-fructose diet (3 g/kg/day) [135].

Despite these findings, fructose is not universally believed to cause NAFLD on its own. It is, however, thought that fructose may favour NAFLD progression on the basis of pre-existing risk factors (e.g., obesity, metabolic syndrome or diabetes) [136,137]. Silbernagel et al. added 150 g/d of fructose or glucose to an otherwise balanced, weight-maintaining diet for 4 weeks in a collective of adolescents. No changes in the amount of hepatic, visceral and subcutaneous fat were found [138].

Fructose is rarely consumed alone, and the effects of fat and fructose have been demonstrated to be additive in the progression of NAFLD. A dietary combination of high fat and fructose resulted in a higher risk of NAFLD and liver fibrosis in adult and rodent studies [128,139].

##### Sugar-Sweetened Beverages and Energy Drinks

Sugar-sweetened beverages or soft drinks (soda, soda pop, cola and tonic) are beverages with a generous amount of high-fructose (corn syrup) or an equally high amount of sucrose [140,141,142]. Over the previous few decades, paediatric consumption of sugar-sweetened beverages, energy drinks and alcohol has increased dramatically [143]. In a study conducted by the European Food Safety Authority (EFSA), 68% of adolescents and 18% of children consumed sugary (energy) drinks on a regular basis [144]. The consumption of soft drinks is associated with paediatric NAFLD and the severity of hepatic fibrosis in a dose-dependent manner [38,87,92,137,145,146].

In adults, consuming more than one soft drink per day (360 mL) was associated with increased risk of the development of metabolic syndrome [147,148]. Table 2 provides a summary of the results for carbohydrates and paediatric NAFLD.

#### 3.2.2. Fat

##### Fat—General Recommendations

The WHO recommends that the total fat intake should not exceed 30% of the total energy intake. The intake of saturated fats should be less than 10% and trans-fat intake should be less than 1% of the total energy intake (see Table 1 for WHO recommendations). Furthermore, a shift in the fat consumption away from saturated fats and trans-fats to unsaturated fats is recommended, with the aim of eliminating industrially produced trans-fats [152]. The American Heart Association (AHA) advises to keep the daily fat intake between 30 and 35 percent of calories for children 2 to 3 years of age and between 25 and 35 percent of calories for children and adolescents 4 to 18 years of age. Most fats should come from sources of polyunsaturated and monounsaturated fatty acids, such as fish, nuts and vegetable oils [153].

##### Fat in the Pathogenesis of Paediatric NAFLD

Consumed fat is mainly TGs that are processed in the intestine and resorbed by enterocytes. Cholesterol is only resorbed at a small percentage from animal products (e.g., eggs and meat). Fatty acids (FA) then reach the liver, where they can either be esterificated or oxidated and accumulated or secreted. FA can be differentiated into saturated FA (SFA), monounsaturated (MUFAs) or polyunsaturated fatty acids (PUFAs) [154,155].

There is an association between fat intake and increased risk of NAFLD. As with carbohydrates, it is more dependent on the type of fat than the total amount consumed [46,156,157]. As previously mentioned, fat and carbohydrates (especially fructose) exert an additive effect when causing NAFLD in the paediatric population [128,139]. A fat-rich diet has been found to induce hepatic steatosis and visceral adiposity [158,159]. Fat-reduced isocaloric diets have also been able to reduce the liver fat content [160].

The quality of fat is not only essential in nutrients, but also on a plasmatic level. Mitchel et al. reported that reducing free fatty acids in the blood flow towards the liver improves steatosis, insulin sensitivity and ALT [75]. In this context, hepatic lipotoxicity is not solely caused by triglyceride accumulation per se, but induced by free fatty acids, free cholesterol and different lipid metabolites (e.g., lysophosphatidylcholine) [161,162].

##### Fat in the Treatment and Prevention of Paediatric NAFLD

Exchanging saturated fatty acids with polyunsaturated (PUFAs) or monounsaturated fatty acids (MUFAs), as advised by the WHO, within isocaloric diets has been shown to improve hepatic fat accumulation [163,164]. Nutritional MUFAs can be found in natural foods, such as milk products, nuts, olive oil, red meat and high-fat fruits, such as olives and avocados [165]. Sunflower oil contains as much as 85% MUFA, and olive oil contains 75% MUFAs. MUFAs reduce the LDL cholesterol, oxidised LDL, TC and TGs levels without decreasing HDL. Replacing carbohydrates and saturated fats with MUFAs reduces glucose and blood pressure and increases HDL in diabetic subjects [165,166,167,168,169]. Nutritional PUFAs can be found in walnuts, sunflower seeds, sesame seeds, peanut butter and peanuts, flaxseed, poppy seeds, avocado oil, olive oil and safflower. The types of PUFA are omega-3 (ω-3) and omega-6 (ω-6) fatty acids. Examples of ω-3 fatty acids are docosahexaenoic acid (DHA), linolenic acid (plant oil) and eicosapentaenoic acid (EPA) (fish oil). Sources of ω-6 fatty acids include eggs, whole-grain foods, walnuts, pumpkin seeds and pine nuts [170]. ω-3 fatty acids are beneficial; however, ω-6 fatty acids should be avoided due to their ability to increase inflammatory processes [171]. Ω-3 fatty acids are ligands of PPAR-α. Low levels of ω-3 PUFAs impair hepatic PPAR-α activity. This decreases mitochondrial lipid β-oxidation and upregulates lipogenic transcriptor factors (e.g., sterol regulatory element-binding protein 1c) [93,172].

In adult NASH patients, not only a high intake of ω-6 fatty acids, but also an abnormal ω-3/ω-6 ratio has been reported [173]. The recommended ω-3/ω-6 ratio is 1:1 to 1:4. A reduction of SFAs in the daily diet and sufficient consumption of ω-3 fatty acids to improve NAFLD in children is advised [70,174,175]. Contrastingly, ω-3 fatty acid supplementation over the course of 6 months in NAFLD children did not improve the ALT levels [132,176,177], while, in another study, the supplementation of ω-3 fatty acids in children with NAFLD over a period of 18 months improved the ALT levels significantly [70].

Diets poor in PUFAs have shown clear association with the development of paediatric NAFLD [178,179]. The increased consumption of saturated long-chain fatty acids (SLCFAs, e.g., palmitic acid, myristic acid and stearic acid), particularly in animal products, causes mitochondrial dysfunction, increased oxidative stress as well as low-grade inflammation [180].

SLCFAs also increase insulin resistance (IR), and the avoidance of SLCFAs benefits IR [181,182].

Furthermore, Anderson et al. stated that the results on fat and paediatric NAFLD remain controversial. Saturated fatty acids (SFAs) have been reported to increase steatosis of the liver proportionally to consumption, as have a high percentage of saturated and a low percentage of polyunsaturated fat consumption in paediatric NAFLD. In a different study, the association between saturated fat intake and NAFLD was reported to be inverse, while in another collective, no association between saturated fat intake and NAFLD could be made [82,83,183,184,185]. Table 3 provides a summary of the results for fat and paediatric NAFLD.

#### 3.2.3. Protein

##### Protein—General Recommendation

The WHO and Food and Agriculture Organization (FAO) recommend a protein intake of 0.9 g/kg/day from 3 to 18 years of age for boys and from 3 to 15 years of age for girls. Between 15 and 18 years of age, the amount decreases slightly for girls to 0.8 g/kg/day (see Table 1 for WHO recommendations) [92].

##### Protein in the Pathogenesis of Paediatric NAFLD

Protein is the least-studied macronutrient concerning pathophysiology and nutritional intervention in paediatric NAFLD [46,186]. From a physiological point of view, protein intake is important for hepatocyte regeneration; it also provides important amino acids for preventing hepatic fat deposition [187].

A number of studies have reported a significantly higher protein intake in patients with NAFLD than in healthy individuals [188,189,190,191], while others found no difference [77,192]. Elevated protein intake may have both beneficial and harmful effects on NAFLD. Some authors report that high protein intake reduces the hepatic fat content [193,194]; other authors, however, suggest that it may increase hepatic fatty acid oxidation and the secretion of fatty acids from the liver [156]. High protein intake may be harmful by influencing gut microbiota fermentation, as it may lead to SCFAs and other intestinal metabolites (e.g., phenols, branched-chain fatty acids and ammonia), thus possibly harming the intestinal barrier’s integrity [195]. A moderate-protein diet (25% of total energy intake) was reported to be without side effects and to reduce the body fat content, similar to a high-protein diet. Hence, authors have proposed a moderate protein diet for NAFLD patients [196].

However, other studies report that malnutrition and protein deficiency also induce hepatic steatosis and NASH [187,197], and are associated with excessive triglyceride storage and NAFLD [198]. A low-protein diet by itself may lead to IR, high blood pressure and lipid abnormalities. These mechanisms are also involved in the development of NAFLD [100,199,200].

##### Protein in the Treatment and Prevention of Paediatric NAFLD

Special diets that are high in protein and low in carbohydrate improve insulin homeostasis, the lipid profile and liver enzymes. Soy protein has been shown to reduce the plasma cholesterol levels and fat deposition in NASH [201]. Table 4 provides a summary of the results for protein and paediatric NAFLD.

### 3.3. Special Issues

#### 3.3.1. Diets

While macronutrients are of great interest, studies tend to compare diets. Results on what the ideal dietary intervention for paediatric NAFLD is remain controversial and, in meta-analysis, no diet has been identified to be superior so far [68]. Long-term compliance with diets is poor and reduced adherence has been associated with a greater increase in liver enzymes [202]. Thus, the management of NAFLD requires multidisciplinary management, including dieticians, psychologists and clinical nurse specialists, to achieve the best results and adherence [203].

##### Low-Caloric Diet

The avoidance of overconsumption and reduction of SFA and free sugar are generally agreed upon components for NAFLD treatment [63,64]. The question of isocaloric vs. hypocaloric diet, however, is complex. Weight loss is considered a main factor beneficial to NAFLD, as obesity and overweightness are common. In meta-analysis, it has been shown that greater weight loss benefitted the improvement of non-invasive NAFLD-markers [68,203]. Weight reduction of 7% to 20% leads to significant improvement in paediatric NAFLD [94,204,205,206,207]. Rapid weight loss is not recommended due to the risk of increasing hepatic inflammation [63].

In a low-calorie diet study combined with weekly nutritional counselling and aerobic exercise (30–45 min/day at least three times per week) by Nobili et al., which was performed for the duration of a year, a significant reduction in BMI resulted in improvements in fasting glucose, insulin levels, serum lipid levels and aminotransferases and hepatic steatosis in sonography. The diet featured 25–30 calories/kg/day, and the constitution of macronutrients was 50–60% carbohydrates, 23–30% fat and 15–20% protein. It also included saturated and unsaturated fatty acids (FA), as well as a 4:1 omega-6/omega-3 ratio [94].

These results were confirmed in a longitudinal randomised controlled trial regarding lifestyle. In a Danish study, 117 children with obesity took part in a 10-week ‘weight loss camp’. After the ten weeks, a year of subsequent dietary and exercise interventions followed. The average weight loss was about 7 kg. Only 71 children completed the trial and showed improvement concerning insulin sensitivity, serum aminotransferases and hepatic steatosis. Over the 12-month follow-up period, only 25% of participants maintained the weight reduction [208].

It has been reported that, as long as weight loss is achieved, a hypocaloric diet, either based on low carbohydrates or low fat, results in comparable reductions in intrahepatic fat [176,209].

Mann et al. reported that the most commonly used hypocaloric diets with low fat contain 25–30 kcal/kg, 50–60% carbohydrates and 23–30% fat (with one-third saturated fat) for nutritional intervention in paediatric NAFLD studies [68,202,203,210,211].

##### Low-Carbohydrate Diets/Low-Glycaemic-Index (GI) Diets

In the FRAGILE study, a specific diet (low fructose/low glycaemic index/low glycaemic load) was investigated in children and adolescents with NAFLD, as well as in healthy individuals. The glycaemic index (GI) is used as a measure of the availability of carbohydrate or the ability of an ingested nutrient to elevate blood glucose. For this purpose, foods are classified as low, medium, or high GI. A GI under 55 is considered low, while 70 and above is considered high GI. Glucose (GI = 100) is high, while sucrose (GI = 65) is considered medium and fructose (GI = 19) is considered low [212,213,214]

It was demonstrated that the absolute fructose intake was strongly associated with plasma aminotransferase, systolic blood pressure, body fat, insulin resistance and serum cholesterol, independent of weight loss [79].

As previously noted, fructose (like sucrose) appears to be a co-factor, but not sole cause, of paediatric NAFLD. As mentioned, evidence supports the fact that there is a significant difference between low-carbohydrate and low-fat diets, as long as both are calorie-restricted diets [215].

Diets low in carbohydrates may be helpful to achieve weight reduction if consumed for a short period (six months). However, weight loss can hardly be maintained after a year by continuing the diet [100].

In order to achieve long-term improvement of hepatic steatosis and improvements in IR, a moderately carbohydrate-restricted diet (CRD) may prove sufficient [76].

This CRD, as presented by Goss et al. for adolescents, aims at minimising the intake of refined carbohydrate sources (e.g., added sugars, high-glycaemic grains and fructose) and to provide ≤25% energy from carbohydrates, 25% energy from protein and ≥50% energy from fat. The intake of saturated fat was limited to <10% total energy/day. They reported that, when compared to a fat-reduced diet, the CRD resulted in significant reductions in weight, BMI, BMI z score, total fat mass and abdominal and leg fat. After the consumption of the CRD, a 32% decrease in the hepatic fat fraction from the average baseline value was achieved. Gross et al. concluded that the dietary recommendation of calorie restriction for weight loss may not be optimal for the reversal of fatty liver in a paediatric population. They concluded that their data suggest that energy restriction may be unnecessary in effectively treating paediatric NAFLD, and significant reductions in the hepatic fat fraction in adolescents can occur in a relatively short period of time (8 weeks) by consuming a CRD [76].

Results are conclusive for low-caloric and low-carbohydrate diets, but are contradictory for isocaloric diets with altered carbohydrate contents and paediatric NAFLD patients. In a collective of 102 children (aged 7–12 years), a low-carbohydrate, high-fat diet was comparable to the control group consuming a standard meal portion [149]. The majority of studies obtained through the literature search focused on carbohydrate reduction in the paediatric population, and only a limited number were found on low-glycaemic index-diets. These diets focused on weight reduction in obesity, but not specifically NAFLD-treatment. Studies comparing low-glycaemic-index diets to a reduced-fat diet in adolescents have shown better improvement of BMI among the low-glycaemic-index groups [149,150,151].

##### Low-Fat Diet

General recommendations concerning nutritional fat have already been discussed (e.g., avoiding SFA and consumption of ω-3 PUFAs). The most frequently used hypocaloric diets are of a low-fat content (25–30 kcal/kg and 23–30% fat (with one-third saturated fat)) for nutritional intervention in paediatric NAFLD-studies [68,202,203,210,211].

When compared with carbohydrate restriction, fat restriction was inferior for the improvement of paediatric NAFLD [76].

##### Mediterranean Diet

The Mediterranean diet (MD) is a diet that has its origins in the states surrounding the Mediterranean Sea. There is no single MD, as all countries have their own variations [216]. The MD consists primarily of vegetables and fresh fruit, olive oil, unrefined cereals and nuts. Fish, white meat and legumes are consumed in moderation. Red meat, processed meats and sweets are rarely consumed. The profile of fatty acids is considered beneficial as there is limited intake of saturated fat and cholesterol and high intake of MUFAs, as well as a balanced ω-3/ω-6 ratio [217,218,219,220].

Due to the beneficial lipid profile, the high content of antioxidants and fibre and low content of simple sugars, the MD is assumed to be anti-inflammatory [163,164,221,222,223].

Therefore, the MD is considered a healthy and simple dietary regimen that is supported by a number of authors to prevent or treat paediatric NAFLD [70,71,72,73,224].

##### Western Diet

The Western diet is hypercaloric and high in fats and simple sugars. This diet is defined by high consumption of fast food, red meat, processed meats, full-fat dairy products, fried potatoes, refined cereals, cakes, biscuits, sweets, soft drinks, sauces and dressings [35]. As a consequence, the postprandial plasma glucose and insulin levels are quickly rising. The western diet is associated with increased hepatic de novo-lipogenesis, hepatic steatosis, insulin resistance, obesity and NAFLD [35,139].

This dietary pattern, if present at the age of 14 years, was shown to be associated with a higher prevalence of NAFLD at the age 17 (independent of sex, frequency of physical activity and sedentary behaviour and family income) [35,207].

##### Ketogenic Diet

In recent years, the ketogenic diet has become more popular. It has been reported that it leads to significant weight loss and reduces the hepatic fat fraction, as well as improves metabolic parameters in patients with obesity. A ketogenic diet is low in carbohydrates and, while being high in fat and/or protein, it is hypocaloric. It is not recommended for children with NAFLD. Maintaining a ketogenic diet over a longer period may aggravate NAFLD and the progression of systemic glucose intolerance, as demonstrated in animal studies [100,221,225].

##### Red Meat

The intake of meat has been associated with glucose intolerance and impaired insulin sensitivity, and may increase the risk of T2D in adults. The metabolic features are also present in NAFLD [226,227]. Grilled meat, predominantly red meat, was found to be associated with NAFLD [228]. In a study by Shi et al., the consumption of red meat was significantly associated with the prevalence of NAFLD, while this was not the case for white meat consumption [229,230].

#### 3.3.2. Dietary Patterns

Dietary patterns concerning frequency, unusual mealtimes and amount of nutrients, as well as the selection of diet are crucial. The quality of food (processed foods and low intake of fruit and vegetables) impacts health and the progress of obesity and NAFLD [231,232,233,234]. It is known that fast eating, irregular meals with multiple feedings and consuming meals on one’s own are potentially harmful in developing eating disorders, obesity and, as a potential consequence, NAFLD. The parents’ function as a positive role model has been discussed [235,236,237,238,239,240,241].

## 4. Discussion

Dietary recommendations for paediatric NAFLD primarily focus on weight loss and support the same regimen as that for treating paediatric obesity [46,242]. General nutritional recommendations focus on a low-caloric healthy diet (low in carbohydrates, fructose, sucrose, HFCS, saturated fatty acids and trans-fatty acids). Currently, there is no specific recommendation for the type of diet and amount of calorie restriction [64]. Low-caloric diets are investigated in studies as either low in carbohydrate and/or low in fat. The results claimed no difference between low fat and low carbohydrate, as long as it was low caloric [176,209]. As obesity is the main risk factor for NAFLD, caloric restriction is associated with weight reduction, thereby benefitting NAFLD (via ALT, ultrasound, MRI and liver biopsy) [10]. Not all authors advocate caloric restriction, but mention the importance of high-quality macronutrients. It is difficult to distinguish the specific effect of the macronutrient per se vs. overall caloric restriction in achieving weight loss. While (iso)caloric restriction appears sensible, hypocaloric diets do not [243]. In striving for weight loss, neglecting macronutrients and needed energy may cause health impairments. This may lead to the disturbance of development, puberty, bone density and potentially worsened NAFLD. It has been reported that prolonged hypocaloric diets (e.g., a ketogenic diet) or rapid weight loss may cause further liver damage [63]. This is also the case with low-carbohydrate diets, as they lead to swift weight loss and improvement of NAFLD over a short period but, if maintained for more than a year, may cause further liver damage [208].

In adults, weight loss of 3–5% can reduce hepatic steatosis and benefit metabolic parameters. NASH can be improved or resolved when more than 7% weight loss is achieved [244,245]. In paediatric NAFLD, the percentage of weight loss necessary to improve NAFLD has been described to be between 7% and 20% [94,204,205,206,207]. Weight reduction should be gradual and not exceed 500 g/week [75]. Moreover, long-term lifestyle changes lead to improved liver histology [246]. Nobili et al. reported that weight loss of >20% over a period of 12 months resulted in improved serum ALT and steatosis in ultrasound. The study did not give any details on the diet used, but provided caloric restriction and exercise advice [207,247]. In a different paediatric study, a mean weight loss of 5 kg (over a 24-month period) resulted in improvements in steatosis, lobular inflammation and liver enzymes [203]. There is no information about the point at which weight loss can be stopped to achieve and maintain hepatic health, or if all patients must aim for normal weight. An important part is that weight loss is not an option for lean patients with NAFLD [94,204,205,206,207].

The Western diet is the diet predominantly associated with NAFLD. It is characterised by the overconsumption of simple sugars and fats. Increased energy intake, low-grade systemic inflammation and metabolic changes associated with this diet drive paediatric NAFLD [35,139]. There has also been a significant association with red, but not white, meat. This may be due to the fact that white meat contains less (saturated) fat. The dietary pattern of people consuming predominantly red meat may also differ (e.g., eating fewer vegetables) from those who consume white meat, but not red meat [229,230]. The Mediterranean diet (MD) is considered a healthy and simple diet, and has been suggested by a variety of authors to prevent and treat paediatric NAFLD. There is no single MD, as all countries around the Mediterranean Sea have their own variety of this diet. The frequency by which certain foods and macronutrients are consumed, as well as the beneficial ratios (e.g., omega fatty acids), are as important as the associated lifestyle. Studies generally list the foods, but the lifestyle associated with MD is more relaxed, with more physical work and exercise, as well as having family meals [217,218,219,220,248,249]. No studies for vegetarian or vegan dietary trials/interventions in paediatric NAFLD were found.

The main focus on single macronutrients and NAFLD in studies is centred around carbohydrates and especially fructose, sucrose and HFCS. While the majority of consumed calories stem from carbohydrates, the total energy and carbohydrate intake are increased in children with NAFLD [86]. There is, however, a focus on free sugars, which should be limited to 10% of the total daily intake and pose a main issue in developing paediatric NAFLD [85]. Several studies showed that following this recommendation benefits the hepatic fat fraction, and improved metabolic parameters as well as liver enzymes could be achieved. Interestingly, while fructose does not cause NAFLD by itself, intake in a refined form is strongly associated with NAFLD [136,137]. When fructose is consumed in its natural form (e.g., fruit), its constituents (flavanols, epicatechin, ascorbate and other antioxidants) may antagonise the negative effects of fructose [115,116]. A relevant issue is the consumption of sugar-sweetened beverages and energy drinks that contain large amounts of sucrose and high-fructose-corn syrup [105]. In studies, even a several-day-long reduction of fructose led to an improvement in liver health [131,135]. In adults, a single 360 mL/d threshold of soft drink has been reported for metabolic syndrome [147,148]. For NAFLD and especially hepatic fibrosis, a dose-dependent association with soft drink consumption exists. In view of this, a significant reduction in fructose, sucrose and HFCS in processed foods, beverages and a general ban on high-fructose corn syrup are advised [38,87,92,137,145,146,250]. The role of fructose in the intestinal microbiome and gut barrier function remains under debate [120,122].

There is an association between the amount of fat consumed and increased NAFLD-risk, but also of the quality of fat. Saturated fatty acids should be avoided in general. As 90% of ingested fat consists of triglycerides (the rest being phospholipids, sphingolipids, cholesterol and the lipophile vitamins A, D, E and K), the main focus should lie on these. In patients’ perception, the role of nutritional cholesterol in metabolic disorders and NAFLD seems overestimated, as only a fraction of ingested cholesterol is absorbed. Patient education should, therefore, aim at MUFAs and PUFAs, as well as the total fat amount and ω-3/ω-6 ratio. Evidence supports the consumption of a diet rich in ω-3-fatty-acids, while ω-6-fatty acids and SLCFAs should be avoided due to their associations with NAFLD. This further advocates the Mediterranean diet as a therapeutic diet of choice for paediatric NAFLD [46,156,157]. Fat and carbohydrates (especially fructose) exert an additive effect in causing NAFLD in the paediatric population [128,139].

There are limited data on protein and paediatric NAFLD and the results are controversial. Although there are reports of significantly higher intake of protein in NAFLD, there is also evidence that diets low in protein are associated with NAFLD. A possible explanation for the higher intake is that a main source of protein is (red) meat, which is also richer in fat and typically associated with a Western diet. Dietary protein is linked to inflammatory processes, influencing the gut microbiome and NAFLD progression. The depletion of protein, on the other hand, may lead to the reduction of muscle mass, inducing gluconeogenesis via arginine from the muscle and thus lowering the capability for glucose resorption and disturbing the metabolic equilibrium [188,189,190,191], while others found no difference [77,192].

## 5. Conclusions

There are key roles of nutrition and especially macronutrients in the pathogenesis of paediatric NAFLD. Fructose, sucrose, HFCS, saturated fatty acids, trans-fatty acids and ω-6 fatty acids are strongly associated with driving paediatric NAFLD [142,171,186,251,252]. On the other hand, fibre, low-glycaemic-index diets, MUFAs and ω-3 fatty acids are beneficial in the prevention and treatment of paediatric NAFLD [253,254,255,256]. There is no evidence supporting children with NAFLD and children with obesity requiring a different nutritional regimen compared with each other and healthy, lean children. Weight normalisation benefits both obesity and NAFLD. As a consequence, the same dietary advice is given to children with NAFLD as to children with obesity. Unfortunately, achieving and maintaining weight loss is very difficult (<10% success over a period of 2 years intervention) [257]. The best results for childhood weight loss have been achieved by multi-disciplinarity family-based behavioural treatment [43]. In meta-analyses, no single diet has been superior in treating paediatric NAFLD [68]. However, the composition of macronutrients in the Mediterranean diet (MD) and certain studies support the MD as suitable for preventing and treating NAFLD [70,71,72,73]. The established recommendations of avoiding saturated fatty acids, trans-fatty acids, simple sugars (fructose, sucrose and HFCS) and a high-caloric Western Diet are supported by the literature [258]. Currently, the optimal proportions of total macronutrients (carbohydrate, fat and protein) in the diet of NAFLD-patients are unknown [74]. The role of dietary protein in paediatric NAFLD remains contradictory [156,193,194]. In addition, at the moment, there are no formal recommendations for the treatment of lean NAFLD [64,245]. Table 5 lists the current recommendations as provided by literature research.

More efforts in preventing paediatric NAFLD by dietary education and the avoidance of harmful macronutrients need to be undertaken. It is clear that combining dietary intervention with physical activity is superior to dietary intervention alone. Further studies on the role of macronutrients and optimal dietary intervention in paediatric NAFLD are needed [259,260,261].

## Figures and Tables

**Table 1 life-12-00839-t001:** WHO macronutrient recommendations in all children.

General WHO Recommendations for Macronutrients in Children [91,92,93,94]	
Carbohydrates	<10% daily of total energy intake
	Reduction of free sugar intake (primarily sucrose and fructose)
Fat	≤30% of total energy intake
	Saturated fats intake: <10% of total energy intake
	Trans fats intake: <1% of total energy intake
Protein	Male (3–18 years old): 0.9 g/kg/day
	Female (3–15 years old): 0.9 g/kg/day
	Female (15–18 years old): 0.8 g/kg/day

**Table 2 life-12-00839-t002:** Summary of the results for carbohydrates and paediatric NAFLD (abbreviations: CHO (carbohydrates), NAFLD (non-alcoholic fatty liver disease), malonyl-CoA (malonyl-Coenzyme-activator)).

Summary Table for Carbohydrates and Paediatric NAFLD	
Pathogenesis:	
	Overconsumption of CHO as a risk factor [95]
	More free fatty acids are synthesised from CHO in paediatric NAFLD [95]
	Additional glucose promotes lipid synthesis [97,98,99]
	Additional glucose impairs mitochondrial protein acetylation and malonyl-CoA [97,98,99]
	Fructose/sucrose: affect gene expression involved in glycolysis and lipogenesis [118]
	Fructose: associated with increased insulin resistance and leptin resistance, and a decrease in adiponectin [69,110,111]
Treatment and prevention:	
	Low glycemic index diet benefits BMI [149,150,151]
	Avoid sugar-sweetened beverages [38,87,92,137,145,146]
	Low-CHO diets may improve weight loss and hepatic steatosis (minus 6–8% in liver fat), insulin resistance, triglyceride and HDL levels
	Avoid processed glucose/fructose [36]
	Fructose: lowering intake may be protective against of NAFLD in children with obesity [131]

**Table 3 life-12-00839-t003:** Summary of the results for fat and paediatric NAFLD (abbreviations: CHO (carbohydrates), NAFLD (non-alcoholic fatty liver disease), MUFAs (mono-unsaturated fatty acids), PUFAs (poly-unsaturated fatty acids), LDL (low-density lipoprotein), TC (total cholesterol), TGs (triglycerides), HDL (high-density lipoprotein), PPAR-α (peroxisome proliferator-activated receptor alpha).

Summary Table for Fat and Paediatric NAFLD	
Pathogenesis:	
	Overconsumption of fat as risk factor [15,46,156]
	Fat and CHO have additive effects in causing NAFLD [128,139]
	Hepatic lipotoxicity, mitochondrial dysfunction, impairment of beta-oxidation and increased oxidative stress, as well as low-grade inflammation [16,17]
	MUFAs reduce LDL cholesterol, oxidised LDL, TC and TGs levels without decreasing HDL [165,166,167,168,169]
	Low levels of ω -3 PUFAs impair hepatic PPAR-α activity [93,172]
Treatment and prevention:	
	Exchanging saturated fatty acids with polyunsaturated (PUFAs) or monounsaturated fatty acids (MUFAs) in isocaloric diets [163,164]
	MUFAs: in milk products, nuts, olive oil, red meat and high-fat fruits, such as olives and avocados [165]
	PUFAs: in walnuts, sunflower seeds, sesame seeds, peanut butter and peanuts, flaxseed, poppy seeds, avocado oil, olive oil and safflower [170]
	ω-3 PUFAs recommended [70,174,175]
	Recommended ω-3/ω-6 ratio is 1:1 to 1:4. [70,174,175]

**Table 4 life-12-00839-t004:** Summary of the results for carbohydrates in Table 3. Summary table for protein and paediatric NAFLD (abbreviations: IR (insulin-resistance), NAFLD (non-alcoholic fatty liver disease)).

Summary Table for Protein and Paediatric NAFLD	
Pathogenesis:	
	Protein intake is important for hepatocyte regeneration [187]
	Protein provides important amino acids for preventing hepatic fat deposition [187]
	Elevated protein intake may have both beneficial and harmful effects on NAFLD [193,194]
	Low-protein diet by itself may lead to IR, high blood pressure and lipid abnormalities [100,199,200]
Treatment and prevention:	
	Moderate-protein diet (25% of total energy intake) was reported to be without side effects and recommended for NAFLD patients [196]
	Results inconclusive [188,189,190,191,193,194]

**Table 5 life-12-00839-t005:** Conclusions from literature research concerning dietary intervention. (abbreviations: MUFAs (mono-unsaturated fatty acids), HFCS (high-fructose corn syrup)).

Recommended	Avoid
ω-3 fatty acids	Fructose, sucrose and HFCS
ω-3/ω-6 ratio of 1:1–1:4	Sugar-sweetened beverages
Monounsaturated fatty acids (MUFAs)	Saturated fatty acids
Dietary Fibre	Trans-fatty acids
Eucaloric diet	ω-6 fatty acids
Low glycemic index diet	(Western-style) Hypercaloric diet

## Data Availability

All articles’ references can be found on the online platform Pubmed.gov.

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
