# Peer review of "The Role of Macronutrients in the Pathogenesis, Prevention and Treatment of Non-Alcoholic Fatty Liver Disease (NAFLD) in the Paediatric Population—A Review"

_life, 2022, doi:10.3390/life12060839_

Round 1
Reviewer 1 Report
There are a number of 1-sentence paragraphs, misspellings, typos, and awkward sentences and paragraphs. Some paragraphs lack a topic sentence or have a long preamble to get to the meat of the matter.
SSB data is all blamed on high fructose corn syrup which does not have significantly more fructose than sucrose sweetened beverages.
The discussion seems to be repetitive - try to write it better/ tighter
It concerns me that high fructose corn syrup is the culprit as data shows there is little difference in the amount of fructose in sucrose and the commonly used form of High Fructose corn syrup. There needs to be more critical evaluation of [papers on low GI and high Gi and no talk about the extreme variablity in the measure and outcomes. There should be more time spent on dietary fiber and its effects.
Line 204 misspelling
Line 216 -words run together should be ‘not only results in hepatic fat reduction, but also ….
Line 495 - is it really inexpensive - it is less than treating the disease byt diet is more costly than some.
Line 557 In adults studies
Line 567 – sentence needs a preposition
Line 570 wordy …high in added sugars and fats
Line 575 – need to suggest that white meat may have less fat and saturated fat, there may not be fewer preservatives as there are poultry sausages etc. It may also be the pattern of consumption with red meats - eg less veg,
Line 578 - studies show that the med diet with lots of fish and veg may not be inexpensive

Reviewer 2 Report
Thomas Pixner and his colleagues consulted the literature and summarized the role of macronutrients in the pathogenesis, prevention and treatment of Non-Alcoholic Fatty Liver Disease (NAFLD) in the pediatric population. Maintaining a eucaloric diet and avoiding saturated fatty acids, simple sugars (mainly fructose) and a high-caloric Western Diet is supported by literature. However, the article has some problems:
- The structure of the article is not a review structure(Materials and Methods,Results), more like research article.
- The article should clearly put forward the title of the two parts: 1. The Role of Macronutrients in the Pathogenesis of NAFLD in the Pediatric Population。 The Role of Macronutrients in the Prevention and Treatment of NAFLD in the Pediatric Population.
- Lack of summary tables (eg :3.2.1. Carbohydrates,2.2. Fat,3.2.3. Protein,3.3.1. Diets.)
- There are too many references, accounting for 17/32 of the total length of the article, and it is suggested to delete some references that prove the same point of view.
Round 2
Reviewer 2 Report
The author has made necessary modify as required. My suggestion is: accept.